# Durability Characterisation of Portland Cement–Carbon Nanotube Nanocomposites

**DOI:** 10.3390/ma13184097

**Published:** 2020-09-15

**Authors:** Alastair J. N. MacLeod, Will P. Gates, Frank Collins

**Affiliations:** Institute for Frontier Materials, Deakin University, Burwood, VIC 3125, Australia; will.gates@deakin.edu.au (W.P.G.); frank.collins@deakin.edu.au (F.C.)

**Keywords:** carbon nanotubes, permeability, X-ray computed microtomography, chloride diffusion, sorptivity, durability

## Abstract

Multiwalled carbon nanotubes have outstanding mechanical properties that, when combined with Portland cement, can provide cementitious composites that could lead to the innovative construction of stronger, lighter, and thinner built infrastructure. This paper addresses a knowledge gap that relates to the durability of CNT–cement composites. The durability to corrosive chloride, uptake of water by sorption, and flow of the permeability of water acting under high water pressure are addressed. Flow simulations were undertaken through segmented 3D pore networks, based on X-ray computed microtomography measurements, the creation of a virtual microstructure, and fluid simulations that were compared with larger-scale samples. The investigation showed decreased water sorptivity of CNT–cement mixtures, indicating improved durability for the cover zone of concrete that is prone to the uptake of water and water-borne corrosives. Chloride diffusion of CNT–cement composites provided up to 63% improvement compared with control samples. The favourable durability bodes well for the construction of long-life CNT-reinforced concrete infrastructure.

## 1. Introduction

With ever-growing societal demands for greater urbanisation and the development of built infrastructure, Portland cement (OPC) is used so ubiquitously that it is the most produced synthetic material on Earth [1]. To reduce its considerable environmental footprint [2], the innovative utilisation of carbon nanotubes (CNTs) in cementitious nanocomposites can provide enhanced strength and durability of the resulting OPC–CNT nanocomposite, leading to greater efficiency of OPC in construction. OPC–CNT nanocomposites can facilitate the production of thinner, lighter, and more durable structural elements, with a reduced requirement for conventional steel reinforcement [1].

The high aspect-ratio fibre morphology of CNTs, combined with their unsurpassed mechanical properties—a tensile strength around 50 times that of steel and 1 TPa elastic modulus [3,4]—has resulted in significant research interest in the development of high-performance OPC–CNT nanocomposites. OPC–CNT nanocomposites have exhibited significant, yet variable, performance enhancements over unreinforced cement paste, including approximately 15–20% compressive strength and 20% flexural strength average enhancements, as well as significantly improved toughness and composite ductility at doses as low as 0.048% by the mass of cement powder [5,6,7,8].

The variability in experimental results has been attributed to both the wide range of CNT doses and dispersion methods employed in different studies [9], as well as the generally poor CNT dispersion in cement mixes due to very strong van der Waals attractions between CNTs (500 eV/µm of tube–tube contact [10]). These forces can be overcome using physical methods to disperse CNTs (e.g., ultrasonication), chemical methods to stabilise the CNT dispersion (e.g., using a surfactant, such as a cement superplasticiser), or a combination. Prior investigations by the authors [11,12] have demonstrated that certain polycarboxylate-based superplasticisers were excellent cement-compatible dispersants for CNTs in cement pastes, when compared to air entrainers, styrene butadiene rubber, calcium naphthalene sulfonate, and lignosulfonate-based superplasticising chemical admixtures.

Previous studies have shown that CNTs may act as nucleation sites for hydration products [13,14], while CNTs have demonstrated an alteration to the hydrated composition of the cementitious nanocomposite [15,16,17], including refinements to the pore network structure [18], indicating an altered hydration behaviour of OPC with the addition of CNTs, and thus enhancement to the mechanical performance of the nanocomposite material. However, in contrast to the extensive study of the mechanical performance of OPC–CNT paste nanocomposites, the durability performance of the material has received far less research attention, despite being an important consideration for the design and construction of concrete structures. For many important deterioration processes, including chloride ingress, it is the structure of the pore network and hardened microstructure that significantly contributes to the durability performance of the material [19], by regulating the relative ease (or difficulty) with which fluids and gases can penetrate into the porous microstructure of cement.

Although limited in number, earlier studies of durability performance of OPC–CNT nanocomposites have reported on the water sorptivity [20,21], water permeability [20] as well as gas [21] and chloride transport [22,23,24], and reinforcement corrosion resistance [25,26], all of which are critically important for applications in reinforced concrete.

OPC–CNT mortar composites were assessed by Han et al (2013) [20] at a single CNT dosage of 0.2 wt.% dispersed using two different surfactant dispersants. The researchers found that the addition of the CNTs decreased the rates of water sorptivity, gas and water permeability by as much as 65%, independent of the surfactant type used. Similarly, in a more recent study [21], Li et al (2020) prepared pastes with superplasticiser dispersed CNTs at 0.08 wt.% dosage and found that the initial water sorptivity was reduced by as much as 55% and the methanol vapour permeability was decreased by as much as 41%. In both studies, the durability enhancements were attributed to microstructure densification and pore refinement due to the admixed CNTs promoting the formation of calcium silicate hydrate.

Studies on the chloride diffusivity through OPC–CNT nanocomposites have predominantly used accelerated methods, such as the rapid chloride ion penetrability test method ASTM C1202 [27] involving the passing of electrical charge through the material. Lee et al (2018) [26] used this approach on CNT-modified mortars with 1 wt.% nanosilica at CNT doses of 0.01–0.07 wt.%. They found an enhanced resistance to passing charge with the addition of the CNTs, above the effect of the added nanosilica, and inferred an optimal dosage of CNTs at 0.03 wt.% to maximise the chloride ion penetration resistance, as measured by a lowered passing charge through the material. In a different study, Sun (2015) [22] added CNTs at mass doses of 0.2–2.0 wt.%, dispersed with sodium dodecyl benzene sulfonate surfactant, to cement mortars. The results showed a maximum charge resistance enhancement of approximately 65% at a CNT dosage of 0.5 wt.%, with some performance deterioration at 1.0 wt.% and 2.0 wt.%, although the high CNT-dosage specimens still outperformed the unmodified control mix. The authors attributed the enhanced chloride ion penetration resistance to a densified cementitious matrix and a refined pore network structure.

However, such rapid proxy methods may not accurately represent the long-term steady state diffusivity of cementitious materials [28], thus experimental measurement of steady-state chloride diffusive transport is necessary to provide an indication of longer-term chloride transport through OPC–CNT nanocomposites. Alafogianni (2019) [24] employed salt spray exposure for 100 days in conjunction with ASTM C1556 [29] to assess the apparent diffusion coefficient in OPC–CNT mortars according to Fick’s law. That study observed at most a 20% reduction in the apparent diffusion coefficient at a CNT dosage of 0.4 wt.% using sodium dodecyl sulphate as a CNT dispersant. Consequently, from the above literature, and despite studies employing different test methods, mix proportions and CNT dispersion approaches, there are conflicting results on the effects of CNTs upon the apparent chloride resistance, and thus durability, of OPC–CNT (mortar) nanocomposites.

Several methods have been used to study the corrosion resistance of embedded reinforcement, which is an important durability parameter for reinforced concrete materials. These include half-cell potential per ASTM C876 [30], measuring electrical potentials attributed to a likelihood of corrosion, and electrochemical polarisation resistance used to calculate the rate of corrosion according to Faraday’s law of electrolysis. A small number of studies on OPC–CNT mortars have demonstrated that, as with the related chloride diffusivity property, low CNT doses were effective in reducing the corrosion rate of embedded steel reinforcement under simulated conditions [26]. Using electrochemical polarisation resistance, Lee et al (2018) [26], in the same study that reported an enhanced passing electrical charge resistance with dispersed CNTs and 1 wt.% nanosilica, found a 55% reduction in the corrosion rate of embedded steel reinforcement with a CNT dose of 0.03 wt.% and 1 wt.% nanosilica compared with only nanosilica. However, the study also found a significant increase in the corrosion rate of 460%–810% at 0.05 wt.% and 0.07 wt.% CNTs, strongly indicating that the corrosion resistance of the CNT-modified pastes was sensitive to CNT dispersion and dosage.

In contrast to Lee et al, Konsta-Gdoutos et al (2017) [25] employed half-cell potential measurement on 28-day OPC–CNT mortars, and found that, over 150 days’ monitoring, mixes with a CNT dosage of 0.1 wt.% remained in a passivated state for 25 days and exhibited a significantly lower total corrosion rate than the mortar dosed with 0.5 wt.% CNTs, which performed comparably to the unmodified reference mortar. The authors correlated the dispersion of CNTs and CNT dosage to the electrochemical corrosion resistance of the cementitious nanocomposite, positing that CNT agglomerations contributed to the formation of localised galvanic couples at the surface of the steel reinforcement, thereby enhancing the corrosion rate in the 0.5 wt.% CNT mortar mix.

However, contrary to the enhanced performance observed by Lee [26] and Konsta-Gdoutos [25], Del Carmen Camacho et al. (2014) [31] found a significant acceleration in the initiation of steel corrosion in mortars with CNT doses of 0.05–0.5 wt.%, by 36 days (23%) at 0.05 wt.%, to 65 days (42%) at 0.5 wt.% compared to unreinforced mortar (155 days), coupled with a higher rate of corrosion in all CNT dosed mixes. The acceleration of corrosion initiation and the higher corrosion rate was attributed to the increased electrical conductivity of the OPC–CNT nanocomposites, together with the relative galvanic potentials of the CNTs and steel reinforcement.

The results of the above studies highlight that CNTs were, in each instance, assumed to be well-dispersed within the cementitious matrix during preparation. Additionally, every study considered the effect of CNTs in concert with the dispersant used—be it a surfactant or a cement-compatible admixture—resulting in potential conflagration of (i) the effects of the CNT dispersant and (ii) CNT dispersion quality with the underlying effects of the CNTs upon the durability of the modified nanocomposite mix.

Consequently, largely due to the contradictory results described above, there remains limited understanding regarding the durability performance of OPC–CNT. There has also been limited investigation of the effects of water sorptivity, permeability, and steady-state chloride diffusion, as well as few comparisons between the effects of (a) dispersed CNTs with and without a chemical dispersant; (b) variation in CNT dosage; and (c) CNT dispersion quality (‘poor’ dispersion quality reflecting agglomerated CNTs) upon the durability performance of OPC–CNT nanocomposites. The purpose of this study was to address these knowledge gaps using a combined experimental and simulated (from X-ray microtomography) approach.

Accordingly, the following effects were investigated experimentally in this study:CNTs with and without a polycarboxylate-based superplasticiser to aid dispersion;CNT dosage rates of 0.05–0.25% by the weight of cement powder;Poor or adequate (i.e., agglomerated or well-dispersed) CNT dispersion quality.

In this study, the water sorptivity, water permeability, and chloride diffusivity, together with the porosity, of five separate nanocomposite mixes were assessed experimentally. Reconstructed microstructures taken from X-ray computed microtomography of the OPC–CNT nanocomposite dosed with 0.1 wt.% CNTs were used to simulate the Darcy flow through OPC–CNT porosity on the microscale and were compared to experimental results.

## 2. Materials and Methods

### 2.1. Materials

Australian type GP (general purpose) Portland cement (conforming to AS3972-2010 [32]) was used as the binder for all paste mixes prepared in this study. The chemical composition for this cement is presented in Table 1. Carbon nanotubes were provided in powder form by Hythane Company LLC (now Eden Innovations Ltd, Perth, WA, Australia) and used as received, as shown in Figure 1. The reported typical diameter range for the multiwalled CNTs used was 25 ± 5 nm, with a carbon content >95% (by mass) and a bulk density of 0.107 g/cm^3^. A commercially available polycarboxylate-based superplasticiser (Viscocrete 6, Sika Australia Pty Ltd. Keysborough, VIC, Australia), classified as a high-range water reducer per AS1478.1 [33], was used both as the CNT dispersant and cement water reducer in this study. In a prior study [12], the authors found that this particular superplasticiser was highly effective at facilitating nanodispersion of agglomerated CNTs into individual nanotubes, as characterised using SEM and UV-vis absorption spectroscopy.

### 2.2. Nanocomposite Paste Preparation

A total of 7 cement paste mixes, including five OPC–CNT nanocomposite mixes, presented in Table 2, were fabricated in this study to investigate the effects of the following upon the hydration behaviour of OPC–CNT nanocomposite pastes:CNTs dispersed with and without the assistance of the superplasticiser—specimens C_10_, N_10_;CNT dose, between 0.05% and 0.25% (by the weight of cement powder)—specimens C_05_, C_10_ and C_25_;Intentionally poorly-dispersed (i.e., agglomerated) CNTs, with the superplasticiser—P_10_.

Reference specimens R_1_ and R_0_ with and without the addition of the cement superplasticiser, respectively, were also prepared for the purpose of comparison with the OPC–CNT specimens. For all OPC–CNT specimens incorporating a superplasticiser, a fixed superplasticiser-to-CNT ratio of 4 was used, following prior studies [16,34] with demonstrated CNT dispersion. A total water (including the superplasticiser)-to-binder ratio (w:b) of 0.4 was used for all mixes.

CNT dispersion and specimen paste mixing was conducted in the following 3-stage procedure:Predispersion—the required dose of the superplasticiser was magnetically stirred with the mix (tap) water for 2 minutes. This step was not employed for specimens R_0_ and N_10_.CNT dispersion—in a fume cupboard, the CNT powder quantity was added to the solution. The mixture was ultrasonicated with ice bath cooling for a total of 20 kJ (10 min) using a Sonics and Materials Vibra-Cell VCX 500 W ultrasonic processor (Sonics & Materials, Newton, CT, USA), equipped with a 19 mm diameter solid cylindrical probe, and operating continuously in energy set point mode. This step was not conducted for specimen P_10_.Paste mixing—the CNT mixture (or mix water, for specimen R_0_, or surfactant mixture, for specimens R_1_ and P_10_) was introduced to a CTE Model 7000 constant speed mixer (Cement Test Equipment, Tulsa, OH, USA); mixing was conducted following the procedure detailed in Section 9.5 of ASTM C1738 [35]. For specimen P_10_ only, CNTs (in a small portion of the mix water) were added to the paste mix immediately following this standard paste mixing procedure, thereby (intentionally) producing an inhomogeneous distribution of CNTs within the nanocomposite.


The paste mixtures were then cast into 50 mm diameter × 100 mm PVC cylinder moulds, vibrated to remove air bubbles, and covered with plastic wrap. After 24 h, the cylinders were demolded and submerged in a lime-saturated bath for saturated curing at 23 ± 2 °C, per Australian Standard AS 1012.8.1 [36]. After 7 and 28 days of curing, cylinders were removed for sectioning and preparation for subsequent durability testing.

### 2.3. Experimental Methods

#### 2.3.1. Water Sorptivity

A modified version of the ASTM test method C1585 [37] was used in this study to determine the unidirectional rate of water absorption of specimens exposed to water. Test specimens prepared for these experiments were 51.0 ± 0.5 mm diameter and 25 ± 1 mm height, compared with 100 ± 6 mm diameter and 50 ± 3 mm height for the standard sized specimens due to a limited amount of nanocomposite material available.

Specimens were sectioned from the cast cylinders using a continuous rim diamond-bladed water-cooled lapidary saw, and conditioned in an environmental chamber for 3 days, at 50 °C and 80% relative humidity. Following preconditioning, per ASTM C1585, the specimens were sealed in individual containers on wire racks for a minimum of 15 days to equilibrate the internal moisture condition of the cement specimens. Just prior to absorption testing on triplicate specimens (procedure detailed in Clause 9 of ASTM C1585 [37]), the average exposed surface area (A_exp_) and specimen height of each specimen was determined from 6 measurements around the circumference. Additionally, the top and vertical surfaces were carefully sealed with plastic wrap and water-impermeable tape. A schematic of the water absorption test set up is shown in Figure 2.

The coefficient of water absorption at time t was calculated according to Equation (1), and delineated into the initial water absorption, S_i_, calculated from the first 6 hours’ absorption, and the secondary absorption, S_s_, from mass measurements (m(t)) after the first day of testing. In each case, the coefficient was determined as the gradient of a least-squares regression fit of the absorption-square root time curve (I(√t)).
(1)$$I\left(t\right)=\frac{m\left(t\right)}{A_{exp}\cdot \rho_{H_{2}O}}=S\sqrt{t}+B$$

#### 2.3.2. Water Permeability

The coefficient of water permeability for cementitious materials under steady-state conditions was characterised experimentally using a flexible-wall permeameter (Global Digital Systems, Hook, Hampshire, UK) test following ASTM D5084 [38]. Biscuit specimens—2.5–6.5 mm thick to reduce the time to achieve steady-state conditions—were sectioned from cylinders, washed to remove cutting debris from the surface, and then immersed in deaired water prior to testing.

Test specimens were installed into a triaxial permeameter cell as shown in Figure 3. Three 3 MPa standard pressure/volume controllers (STDDPC V2) from GDS Instruments (Global Digital Systems, Hook, Hampshire, UK) were used to provide and record the confining, input, and output water pressures of 1600 kPa, 1510 kPa, and 10 kPa, respectively, corresponding to a pressure gradient of 1500 kPa across the specimens. Tests were conducted over 4–5 days (until steady-state conditions were achieved—i.e., approximately equivalent inflow and outflow rates) with a recording interval of 60 s.

The water permeability coefficient, K (m/s), was subsequently calculated according to Darcy’s law [39], Equation (2), with the average steady-state flow rate, Q (m^3^/s), from the least-squares regression of the measured volume change with time of the steady-state flow, under a pressure gradient of 1500 kPa across the specimens.
(2)$$\frac{Q}{A}=-\frac{k}{\mu }\frac{\Delta P}{L}=-K\frac{\Delta P}{L}$$

#### 2.3.3. Steady-State Chloride Diffusivity

A steady-state chloride ion diffusion test was conducted on sectioned 28 day-old biscuit specimens R_1_, C_05_, C_10_, P_10_, and C_25_ (mean thicknesses of 2.1–4.5 mm) over a period of 9 months, using the set up illustrated schematically in Figure 4. After sectioning and rinsing with distilled water to remove cutting detritus, duplicate specimens were carefully installed into the apparatus, which was then gently clamped and sealed with water- and chloride-impermeable tape to ensure a tight seal. Poured into the reservoirs on either side of the test specimens was 2.5 L of 0.6 M NaOH sink solution, and 0.6 M NaOH with the 0.6 M NaCl test solution.

Periodically, 12 mL aliquots of the sink solution (0.6 M NaOH) were sampled for subsequent chloride content analysis, which was conducted using potentiometric titration of 0.05 M silver nitrate (AgNO_3_) solution with a Metrohm 848 Titrino Plus autotitrator (Metrohm Australia, Mitcham, VIC, Australia), equipped with a platinum electrode and a 20 mL burette (with a detection limit of 2 ppm). Of the aliquots 6 mL samples were used for each titration measurement, and the average value of two measurements was used for each specimen at each testing point.

The steady-state chloride diffusion coefficient for each mix (mean of the duplicate specimens) was then calculated from Fick’s 1st law of diffusion [40], shown in Equation (3), from the linear LSR fit for the chloride content (C_sink_) over the final 3 months’ (i.e., 6–9 months) measurement. The formation of Friedel’s salt (from monosulphate) and the chemisorption of chloride to calcium silicate hydrate (C-S-H), the principal mechanisms of chloride binding [41], were not considered. Error bars presented in results reflect ± 1 standard deviation associated with measurement and sampling estimated uncertainties.
(3)$$J_{x}=-D_{x}\frac{\delta C}{\delta x}=\frac{V}{A}\cdot \frac{\delta C_{sink}}{\delta t}=\frac{D_{app}}{l}\left(C_{source}-C_{sink}\right)$$

#### 2.3.4. Nitrogen Adsorption Porosimetry

Characterisation of pore size distribution in 7-day specimens was performed using a BELsorp mini II gas adsorption porosimeter (MicrotracBEL Corp., Nagoya, Aichi, Japan), with nitrogen as the adsorbing gas. Nitrogen adsorption porosimetry was employed in this study to characterise the changes to the pore network structure in the size range of the CNTs (i.e. below 100 nm), and to overcome some of the limitations of mercury intrusion porosimetry, such as the ink bottle pore effect, that may make mercury porosimetry an unsuitable method for pore network characterisation in cementitious materials [42].

Porosimetry specimens were cut from sectioned cylinders into 1–3 mm diameter cubes, immersed in isopropanol for 3 days, then vacuum dried for a minimum of 5 days before testing. Approximately 0.2 g of dried material was used for each of the duplicate test specimens. For each experiment 31-point adsorption and 43-point desorption isotherms with dead volume correction were gathered. The Barrett, Joyner, and Halenda (BJH) method [43,44] (with a standard silica t-curve) was used to calculate the pore size distribution using capillary condensation of the desorption stage, corresponding to pore diameters in the range of 3–190 nm.

#### 2.3.5. X-ray Computed Microtomography and Porosity Segmentation

The examination and subsequent segmentation of the microstructure of specimens for fluid flow simulations was performed according to a process detailed in a companion study previously reported by the authors [16]. Cropped 500 µm × 500 µm × 500 µm volumes (resolution of 0.9–1.3 µm) were produced from the X-ray computed microtomography (XR-µCT) experimental datasets investigated in this study. Here, only the threshold differentiating between the porosity and solid fractions of the microstructure was considered, using the tangent image intensity histogram method to segment the solid and pore phases.

### 2.4. Simulations

#### 2.4.1. Simulated Cement Hydration and Microstructural Development

The Virtual Cement and Concrete Testing Laboratory software (VCCTL, Version 9.5, National Institute of Standards and Technology, Gaithersburg, MD, USA) [45,46,47,48] was employed to simulate the 3D microstructural development of a virtual cement with similar composition and curing conditions to the experimental reference paste R_1_ used in this research. The virtual cement, at simulated ages of 7 and 28 days of hydration, was then segmented and characterised within the Avizo 9.0.1 software (Version 9.0.1, Thermo Fisher Scientific Australia, Scoresby, VIC, Australia) environment for comparison with the experimental results.

The simulation of the virtual cement was conducted in three stages: definition of the cement and initial conditions; hydration simulation; and segmentation and analysis. The first requirement was the selection of appropriate virtual cement from the available cement powders (each characterised in detail by the cement and concrete reference laboratory, CCRL) in the VCCTL software, together with the definition of the microstructural model. The CCRL proficiency sample Cement 116 was selected for the hydration simulation. Table 3 shows a comparison with the calculated Bogue proportions for type GP cement (calculated from the compositional information presented in Table 1). A realistic particle shape model set (“Cement 140”) was used for the definition of the cement particle shapes. To roughly approximate the dispersing effect of a superplasticiser on cement grains in the simulated hydration within the virtual microstructure, a 1 µm separation (minimum resolution of the simulation) was imposed between neighbouring cement grains. A water-to-cement ratio of 0.4 was used in setting up the simulated volume. A volume of 200 pixels × 200 pixels × 200 pixels (corresponding to 200 µm × 200 µm × 200 µm) was chosen for the size of the model.

Following the creation of the model microstructure prior to hydration, the model was simulated under saturated hydration conditions (i.e., no desiccation as hydration progresses) at 25 °C up to 28 days of hydration. The cumulative heat of hydration calorimetric profile for specimen R_1_ was used to fit the simulated hydration progress more accurately to the experimental data.

After simulated hydration, the virtual cement at 7 and 28 days of simulated hydration was segmented into the solid and porous phases based upon the assigned phase colour within Avizo 9.0.1 software (Version 9.0.1, Thermo Fisher Scientific Australia, Scoresby, VIC, Australia) environment for subsequent fluid flow simulation.

#### 2.4.2. Fluid Flow Simulation

The numerical simulation of fluid flow through the 3D pore networks of reconstructed XR-µCT and VCCTL microstructures was conducted within the Avizo 9.0.1 software environment [49], using the XLab Hydro extension module [50], in order to numerically estimate the absolute permeability of the reconstructed microstructures for subsequent comparison with the permeation experiments.
(4)$$\vec{\nabla }\cdot \vec{V}=0$$
(5)$$\mu \nabla^{2}\vec{V}-\vec{\nabla }P=\vec{0}$$

The software numerically estimates the absolute permeability of a connected, segmented region of interest (such as the pore network) by solving the simplified Navier–Stokes equations, in Equations (4) and (5), assuming laminar and steady-state flow of an incompressible, Newtonian fluid. In the Navier–Stokes equations, V is the fluid velocity, µ is the dynamic viscosity of the flowing fluid (8.9 × 10^–4^ Pa.s at 25 °C for water), and P is the pressure of the flowing fluid. These equations are solved numerically for flow in one of the three cardinal directions (i.e., X, Y, or Z axes) using the finite volume method. The coefficient of absolute permeability, k (m^2^), is then estimated by applying Darcy’s Law [39].

Simulations were conducted on 100 μm × 100 μm × 100 µm cube volumes of interest (VOI), previously segmented between pores and solid materials. Selected regions contained connected pore networks (i.e., pore regions connected to the edges of the VOI in at least one of the three cardinal directions), but, due to software restrictions, only 6-voxel connectivity was considered for the connected pore network.

For each specimen tested (XR-µCT specimens R_1_ 7 days, C_10_ 7 days, and C_10_ 28 days; VCCTL specimen V_1_ aged 7 days and V_1_ aged 28 days), simulations were conducted on a minimum of six separate regions. The test arrangement, including imposed boundary conditions, is illustrated in Figure 5. In addition to the estimates of the absolute permeability coefficient, and the associated standard deviation of the result, each simulation yielded pressure gradient field and flow velocity streamlines that provided qualitative information on the movement of fluid within the reconstructed microstructures.

To provide as close an approximation as possible to the macroscale experimental permeability tests conducted on OPC and OPC–CNT specimens (described in Section 2.3.2), the following conditions were applied to the simulations:Separate simulations were conducted in each of the three cardinal directions in turn for each of the VOI investigated. However, due to the limited 6-voxel pore connectivity applied for the simulations, each simulation was checked to ensure that the result corresponded to a continuous, connected pore network.A dynamic viscosity of 8.9 × 10^−4^ Pa.s, corresponding to water at 25 °C.As a result of the limited pore voxel connectivity used in the simulations, a refining coefficient of 2 was applied to intentionally oversample the VOI, to increase the precision of the evaluation of the unknowns, particularly for narrow pore throats.An inlet pressure of 1.51 MPa and an outlet pressure of 0.01 MPa were applied to each simulation.

## 3. Results and Discussion

### 3.1. Absorption—Water Sorptivity

The rates of water absorption—S_i_ for the initial rate (up to 6 h) and Ss for the secondary rate (after 24 h)—for all specimens after 7 and 28 days of curing are presented in Figure 6. Across all specimens (except C_25_ aged 28 days, which was damaged during sectioning), the coefficient of initial water absorption, S_i_, was smaller at 28 days of age than 7, due to the slow, ongoing hydration reactions, which refine the capillary porosity within the cement, resulting in an overall increase in the resistance of the material to water capillary uptake in unsaturated exposure conditions with age.

Comparing the results for specimens R_0_ and N_10_, the introduction of 0.1% CNTs (without the use of superplasticisers) imparted a 12% increase in initial sorptivity (i.e., increased initial capillary uptake, S_i_) at 7 days of age compared to the unmodified cement. However, at 28 days, although the expected ongoing hydration reactions in the reference specimen R_0_ resulted in a small decrease in the sorptivity coefficient, the nanocomposite N_10_ exhibited a dramatic reduction in the sorptivity (32% smaller than for the reference at the same age), indicative of a significant refinement to the connected capillary pore network within the nanocomposite specimen compared to the unmodified reference paste. A companion study [16] on the effects of CNTs upon the microstructural development of the OPC–CNT nanocomposite demonstrated quantitatively that, as the nanocomposite aged, specimen N_10_ showed an increase in the relative proportion of hydration products including C-S-H. This resulted in a further refinement of the pore network—compared to that observed in specimen R_0_—and led to the 45% reduction in the rate of initial water absorption from the 7-day specimen. A similar but significantly smaller trend was observed for the two specimens in the rates of secondary absorption, S_s_.

These effects were also observed in specimens with the addition of superplasticiser-dispersed CNTs—C_05_, C_10_, and C_25_ (but not at 28 days for C_25_, likely due to damage during sectioning), as well as P_10_ with poorly-dispersed CNTs—when compared to reference specimen R_1_. Specimens C_10_ and P_10_ exhibited a similar overall increasing trend in the rates of water absorption—both initial and secondary—at 7 and 28 days. However, the magnitude of the difference relative to the reference mix R_1_ was larger in the poorly dispersed specimen, with a 46% higher S_s_ at 7 days of hydration (compared to 7% for C_10_, with the same dosage of CNTs), and 153% for P_10_ at 28 days of hydration compared to 118% for C_10_. This finding was significant, as, to the authors’ best knowledge, for the first time, it demonstrated that poorly dispersed or agglomerated CNTs result in a deterioration of the durability performance of the nanocomposite. Here, the effect was attributed to a more connected surface pore network from the agglomerated CNTs contributing to an overall increased porosity of the nanocomposite, discussed further in Section 3.4, and thus leading to the observed deterioration in capillary absorption resistance.

### 3.2. Permeation—Water Permeability

Table 4 presents the 7-day water permeability coefficients determined on the specimens tested. The introduction of CNTs dispersed without the assistance of a superplasticiser in specimen N_10_ had a significant effect in reducing the relative water permeability of the nanocomposite relative to the reference specimen R_0_, closely reflecting the observed relationship in the unsaturated capillary absorption at 28 days. Together, these results strongly imply a less continuous pore network at the gel and fine capillary pore scale, leading to a substantial reduction in the relative water permeability of the material, from cement hydration reactions producing (amongst other products) C-S-H and calcium hydroxide. This effect is illustrated in the C-S-H formations surrounding dispersed CNTs shown in Figure 7. This finding is supported by microstructural characterisation conducted on the same material reported in a separate study [16].

A similar effect upon the water permeability coefficient was observed with the addition of the superplasticiser in reference specimen R_1_, and likewise reflects a more discontinuous pore network within the material resulting from better compaction from enhanced fluidity of the mix in its plastic state. Additionally, the effects of swelling of the C-S-H under the hydrostatic pressure [51] could have contributed to a relative reduction in the permeability of the paste concomitant with the more discontinuous pore network. However, unlike R_1_, the addition of superplasticiser dispersed CNTs in specimens C_05_, C_10_, and C_25_ showed a marked lower resistance to water permeation, largely counteracting the effects observed in R_1_ with the addition of the superplasticiser alone, and the trend was similar to results observed for secondary water absorption in Section 3.1.

This indicates that, contrary to the discontinuous pore network effects of the superplasticiser, the CNTs at 0.05, 0.1, and 0.25 wt.% contributed to a more connected pore network relative to R_1_, with CNTs, and particularly localised aggregations, connecting between porous regions of the microstructure. Further evidence is discussed in the analysis of the pore size distribution results in Section 3.5.

The influence of extensive CNT agglomeration in specimen P_10_ was found to result in only a 6% increase in the permeability coefficient compared to the well-dispersed specimen C_10_, showing that the extensive CNT agglomeration within specimen P_10_ did not dominate the connected porosity (and thus the permeability) within the poorly dispersed specimen at 7 days of hydration.

### 3.3. Fluid Flow Simulations in Reconstructed Microstructures

The microstructures used in the 3D fluid flow simulations were gathered from two sources: (a) reconstructed microstructures from X-ray microtomography of experimental specimens R_1_ and C_10_ (Section 2.3.5), and (b), as shown in Figure 8, virtual microstructures from cement hydration simulations (Section 2.4.1), specimen V_1_ (in this study, not simulating the addition of CNTs, rather as a virtual, reference comparison mix to the nanocomposite). Table 5 summarises the hydraulic conductivity results from the simulations conducted on the tested subvolumes in the three cardinal directions.

An initial quantitative magnitude comparison between the simulated results (on the order of 10^−8^ m/s) and the experimental (10^−12^–10^−13^ m/s) indicates poor correlation between the simulations and experimental data. However, careful consideration of the following influencing factors may be considered to account for some of the quantitative discrepancies between the experiments and simulated results:Simulated flow was restricted to only the percolating pore network(s) within the segmented microstructures (i.e., no flow through the segmented C-S-H).The reconstructed microstructures (including the virtual samples V_1_ 7 days and 28 days) were limited to a nominal resolution of around 1 µm. This factor, more than any other, would significantly alter the permeability of the reconstructed volumes compared to the experimental samples.The experimental test results were influenced by factors including C-S-H swelling, blockages and specimen microcracking, as discussed in Section 3.2.

Anisotropy in the hydraulic conductivities in the three cardinal directions could indicate that CNTs (or, more precisely given the resolution of the reconstructed volumes, CNT agglomerations), with their highly directional morphology, would cause directional pore development within the material, an outcome consistent with the continued ongoing hydration process observed in the nanocomposites. However, the results across all specimens, except C_10_ 7 days, indicated no obvious directional behaviour. For C_10_ 7 days, the marginal anisotropy observed in the reported directional conductivities was a result of CNT agglomerations segmented as porosity. However, further investigation is required to determine the precise behaviour of fluids, particularly water, within and through agglomerated CNT structures in cements. Comparing the average results for specimens R_1_ 7 days and C_10_ 7 days, the nanocomposite simulated permeability was 3.5 times (less than an order of magnitude) that of the reference material.

Further, visualisations of the flow paths through the microstructures, exemplified in Figure 9, highlight the tortuous routes and relative spatial relationships between regions of higher flow and solid materials. In each figure, the velocity of the simulated flow path is represented by a red-blue (high to low velocity) colour gradient, while the number of neighbouring flow lines is indicative of the flow volume.

A beneficial effect was observed upon the permeability performance as the nanocomposite aged; the 28-day specimen C_10_ exhibited a 56% lower average hydraulic conductivity than the 7-day C_10_ specimen. Conversely, a comparison between the virtual samples V_1_ 7 days and 28 days showed a 15% reduction in the hydraulic conductivity over the same time increment.

The virtual specimens (V_1_) presented a single sample microstructure after 7- and 28-day periods of simulated hydration in the VCCTL software. With a pixel resolution of 1 µm, the resulting microstructure (cube of 200 µm side lengths) had a comparable spatial resolution to the reconstructed experimental microstructures gathered using XR-µCT.

In this study, the virtual microstructures were used to gauge their potential for future use with simulating OPC–CNT materials. However, although progress has been made in the formation of microstructural hydration simulations of cementitious and concrete materials, results for reference materials (V_1_ and R_1_) do not yet reflect experimental observations. Comparing the simulated permeability results for the experimental and virtual microstructures, shown in Figure 9 and Figure 10 respectively in addition to Table 5, the virtual microstructure consistently overestimated the permeability flow (at 1 µm resolution) by approximately 3–5 times at both 7 and 28 days.

This discrepancy is a consequence of the substantially larger proportion of porosity present in the virtual specimen, resulting in a lower pore network tortuosity and higher permeability. Accordingly, this shows that further development of the virtual hydration and microstructural models (at the mesoscale, corresponding to C-S-H aggregations) is necessary prior to the incorporation of CNT-type structures (even as agglomerations) into the virtual systems.

### 3.4. Diffusion—Chloride Diffusivity

The apparent chloride diffusion coefficients (D_app_) for specimens R_1_, C_05_, C_10_, C_25_, and P_10_ (all cured for 28 days prior to testing for a total of 9 months) are presented in Figure 11. A comparison of the results reveals that the addition of well-dispersed CNTs up to a dosage of 0.1 wt.% reduced the apparent chloride diffusion coefficient of the cementitious nanocomposite by as much as 63% compared with the reference mix R_1_, and the poorly dispersed CNT mix, P_10_, also showed a 34% reduction in the mean diffusion coefficient, but also significantly more sample variability.

It is proposed that agglomeration of the CNTs within the cementitious microstructure, with a concomitant inhomogeneous pore network structure through the specimens (discussed further in Section 3.4), was a principal cause of the poorer chloride diffusion resistance in the P_10_ mix compared to that of the well-dispersed mix with the same dosage of CNTs, C_10_.

In contrast to the lower dosage mixes, specimen C_25_, with 0.25 wt.% CNTs, exhibited only a 1% (within the estimated uncertainty for the experiment) lower mean resistance to chloride diffusion compared to the reference mix. With the higher CNT dosage loading within the nanocomposite, it is hypothesised, and supported by previous research quantitatively characterising the size and spatial distribution of porosity (and CNTs) within the material [16], that there was a greater prevalence of randomly distributed small, localised CNT agglomerations, as well as closer average spacing of CNTs throughout the nanocomposite microstructure, thereby leading to additional diffusive pathways for chloride ions to transport through the material.

The relative performance of mixes R_1_ and C_05_ upon the service life of a hypothetical steel reinforced material, under assumed exposure conditions of 3.5% surface chloride and a critical chloride threshold of 1% (by the weight of cement) at the surface of the embedded reinforcement, was calculated per Fick’s law and is shown in Figure 12. It demonstrates that the reduction in mean D_app_ would translate into an average 268% increase (range of 150–600%) in the time for chloride to penetrate the cover to reinforcement.

### 3.5. Porosity

Incremental pore size distributions between pore diameters of 2.5–200 nm for specimens aged 7 days are presented in Figure 13, consisting of a peak at 20–30 nm, corresponding to the main volume of medium capillary porosity within the material, and a sharp peak at 3.8 nm diameter, associated with the nitrogen-accessible gel capillary porosity of the low-density C-S-H hydration product [44,52].

The addition of 0.1 wt.% CNTs (but no superplasticiser) in specimen N_10_ showed two important shifts in the PSD compared to the reference mix R_0_. The first was a shift towards a smaller peak medium capillary porosity of 25 nm compared to approximately 35 nm for the reference mix and was similar to the difference between R_0_ and R_1_ (reference with the superplasticiser). This is indicative of a refinement of the pore network resulting from a densification of the microstructure with formation of hydration products, specifically C-S-H (and calcium hydroxide) in the presence of the CNTs. The second was a 35% reduction in the nitrogen-accessible C-S-H peak at 3.8 nm, indicating a development of more highly ordered and higher density C-S-H in the presence of the CNTs as opposed to low-density, porous C-S-H structures without CNTs. In contrast, the reference mix R_1_ showed a 70% increase in the C-S-H peak, implying the formation of more nitrogen-accessible C-S-H, which, as mentioned above, was coupled with a reduction in the peak of medium capillary pores (20 nm approximately). Accordingly, although both N_10_ and R_1_ (relative to R_0_) showed a densification of the pore network structure, the structure of the hydrated phase formed was higher density in the CNT-dosed nanocomposite compared to the superplasticiser-modified cement paste.

The superplasticiser-modified CNT mixes C_05_ and C_10_ showed similar effects upon the nitrogen-accessible C-S-H porosity to that N_10_–C_10_ and N_10_, both with 0.1 wt.% CNTs, were virtually identical. Thus, it appears that the structure of the hydration products like C-S-H around the well-dispersed CNTs governs the measured nitrogen-accessible porosity in the pore diameter range below 10 nm. However, unlike N_10_, C_05_, and C_10_ both showed a larger peak medium capillary pore diameter than the reference mix R_1_, at 28 nm and 44 nm, respectively. This result indicates that the C_10_ mix had some localised CNT agglomerations within its microstructure—due to incomplete dispersion during the fabrication stage—that resulted in an increase in the medium capillary porosity peak and led to a more connected pore network and a concomitant greater permeability.

In contrast, although the 0.1 wt.% CNT mix with deliberately extensively agglomerated CNTs, P_10_, exhibited a peak medium capillary pore diameter around 25 nm, it showed a 185% greater pore volume in the gel capillary pores (3.8 nm diameter) compared to C_10_, implying the formation of 2.85 times the quantity of nitrogen-accessible (or low-density) C-S-H products. This demonstrates that large, extensive CNT agglomerations significantly alter the hydration product formations that develop in the cementitious nanocomposite, which is further supported by X-ray computed microtomography of CNT agglomerations in several studies [16,53].

## 4. Conclusions

This study experimentally assessed the water sorptivity, water permeability, chloride diffusivity, and porosity of CNT-reinforced cement pastes: (i) CNTs dispersed with and without a polycarboxylate-based superplasticiser dispersant; (ii) varying CNT dosage between 0.05 to 0.25 wt.%; and (iii) CNT dispersion quality (i.e., adequate or poor dispersion).

Results showed that the initial water sorptivity of a 28-day cement paste with 0.1 wt.% CNTs (without a superplasticiser) was up to 32% lower than the reference cement paste. However, poorly-dispersed 0.1 wt.% CNTs (with a superplasticiser) exhibited a deterioration in water sorptivity resistance of up to 153%, resulting from a more connected surface pore network from the agglomerated CNTs, which was supported by porosimetry characterization.

Water permeability coefficients under a 1.5 MPa pressure gradient for the superplasticiser-dispersed nanocomposite pastes with 0.05, 0.1, and 0.25 wt.% CNTs were 3–5 times greater than the reference paste mix R_1_. Similarly, simulated fluid flow in specimens R_1_ and C_10_ showed a 3.4 times greater mean permeability through the reconstructed OPC–CNT microstructure. However, the virtual microstructures from hydration simulation—assessed in comparison to the OPC–CNT materials—consistently overestimated the conductivity and pore network connectivity compared with the reference paste R_1_, indicating that further refinement of simulated microstructures will be required to more accurately reflect the discontinuous pore network structure for future permeability simulations.

Steady-state chloride diffusion tests demonstrated that the chloride diffusion coefficient was reduced by up to 63% with the addition of superplasticiser-dispersed CNTs at a dosage of 0.05–0.1 wt.%, increasing service life by as much as 2.7 times. 0.25 wt.% CNTs did not provide an enhanced resistance to chloride diffusion compared to the reference cement paste, R_1_, implying an optimum CNT dosage for enhanced chloride diffusion resistance of 0.05–0.1 wt.%.

Overall, the results showed that CNT dosage was more influential over the durability characteristics of the cementitious nanocomposite than CNT dispersion. Further, at doses of 0.05–0.1 wt.%, CNTs enhanced the resistance to chloride diffusion of the material, with important implications for service life. The favourable chloride resistance bodes well for the future construction of long-life CNT-reinforced concrete durable infrastructure.

## Figures and Tables

**Figure 1 materials-13-04097-f001:**
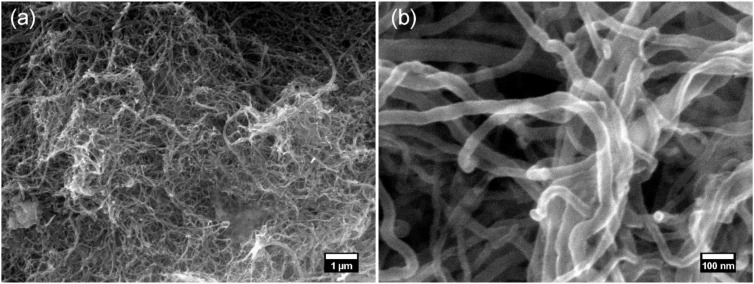
Secondary electron micrographs (JEOL 7001F FEGSEM at 15 keV) of the as-delivered carbon nanotubes (CNTs). (**a**) A 10,000× magnification showing the heavily agglomerated CNTs. (**b**) A 100,000× magnification showing the morphology of the entangled CNTs prior to dispersion.

**Figure 2 materials-13-04097-f002:**
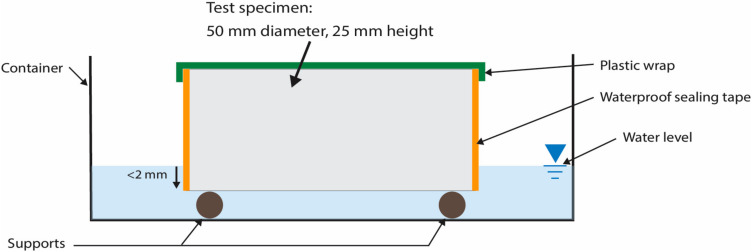
Schematic of the experimental setup for sorptivity testing, per the modified ASTM C1585 test method.

**Figure 3 materials-13-04097-f003:**
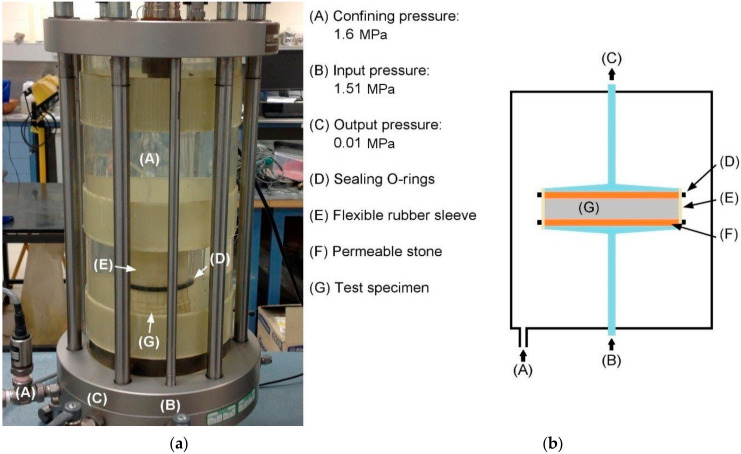
Experimental permeameter cell used in this study. Annotated photograph of the setup (**a**) and (**b**) a basic schematic representation of the equipment.

**Figure 4 materials-13-04097-f004:**
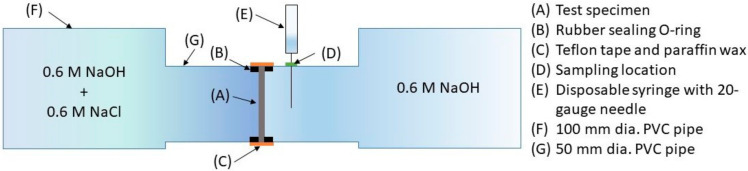
Schematic illustration of the chloride diffusion testing arrangement.

**Figure 5 materials-13-04097-f005:**
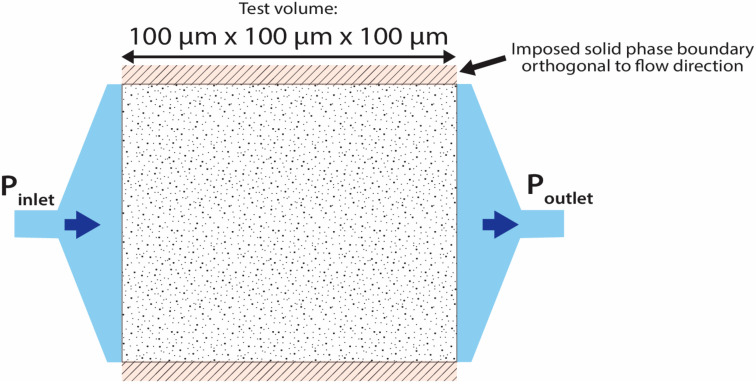
Schematic of permeability simulation setup.

**Figure 6 materials-13-04097-f006:**
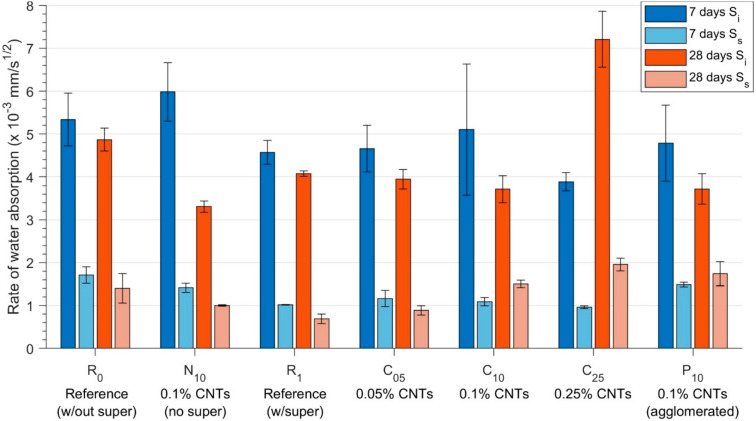
Calculated coefficients of initial (S_i_) and secondary (S_s_) water absorption (sorptivity, mm/s^1/2^) for specimens at 7 days and 28 days. Error bars show ± 1 standard deviation of results.

**Figure 7 materials-13-04097-f007:**
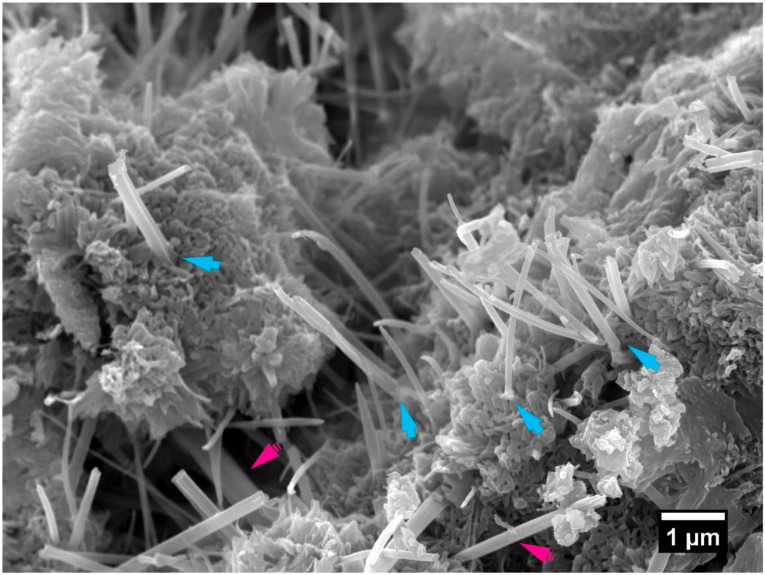
A secondary electron micrograph of fracture surface of specimen C_10_, 7 days, showing CNTs embedded within C-S-H formations (blue arrows) within the hydrated cementitious microstructure. Ettringite structures are indicated by magenta arrows.

**Figure 8 materials-13-04097-f008:**
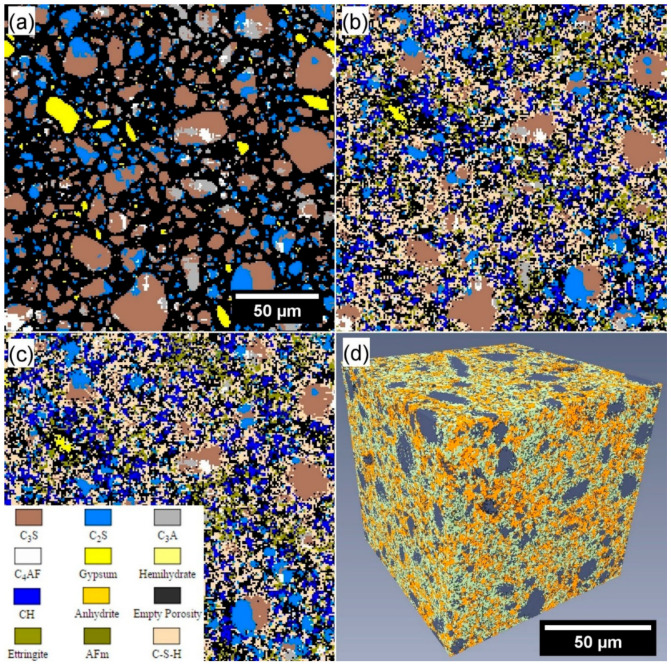
(**a**–**c**) Colour-coded VCCTL microstructural slices after (**a**) 0 hours, (**b**) 7 days, and (**c**) 28 days of simulated hydration of the virtual cement V_1_. (**d**) Reconstructed volume rendering of the segmented microstructure at 28 days of hydration, within the Avizo software environment.

**Figure 9 materials-13-04097-f009:**
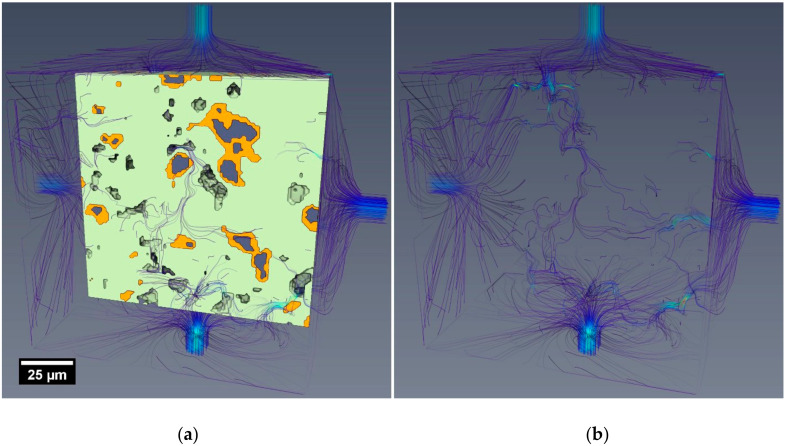
(**a**) Volume rendering of microstructure overlaid with simulated flow visualization. (**b**) Flow visualisations in X and Y directions (colour indicates velocity) for a 100 µm cube (VOI) from the XR-µCT scan of specimen C_10_ 7 days.

**Figure 10 materials-13-04097-f010:**
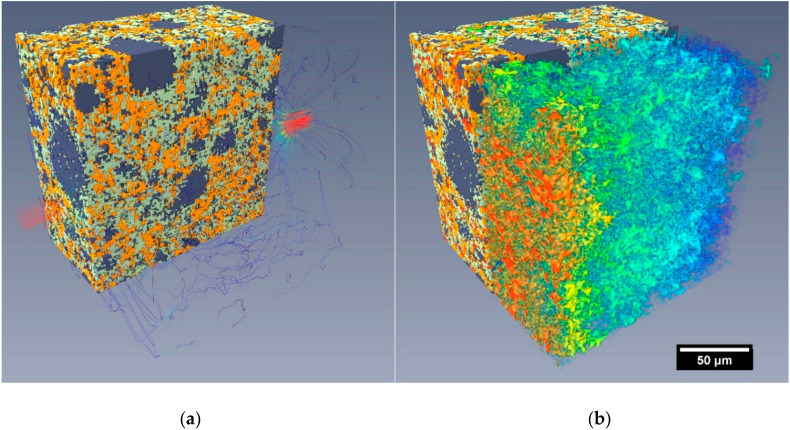
Volume rendering of the microstructure of VCCTL specimen V_1_ 7 days overlaid with (**a**) simulated flow visualisation in the X-direction (red to blue colours indicating high to low relative velocity). (**b**) The corresponding pressure field, with colour indicating the relative pressure (red high pressure, 1.51 MPa, to blue low pressure, 0.01 MPa).

**Figure 11 materials-13-04097-f011:**
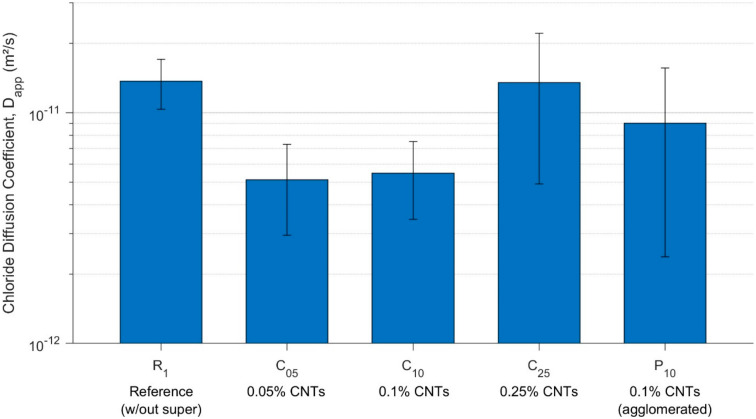
Comparison of the calculated coefficients of chloride diffusivity for specimens. Error bars show ±1 s.d. of results across specimens (including estimated experimental uncertainties).

**Figure 12 materials-13-04097-f012:**
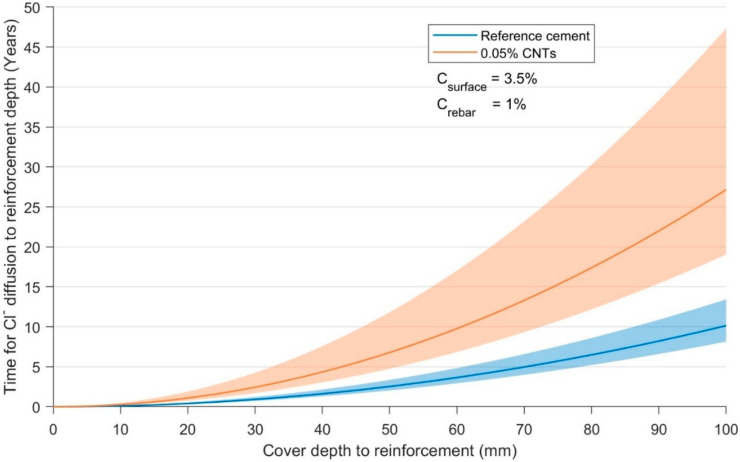
Calculated initiation time of corrosion as the function of cover depth to reinforcement, comparing apparent chloride diffusion coefficients for specimens R_1_ and C_05_, assuming surface chloride concentration of 3.5% and a critical chloride threshold of 1% (by the weight of cement). Shaded regions denote values ±1 s.d. of apparent diffusion coefficient.

**Figure 13 materials-13-04097-f013:**
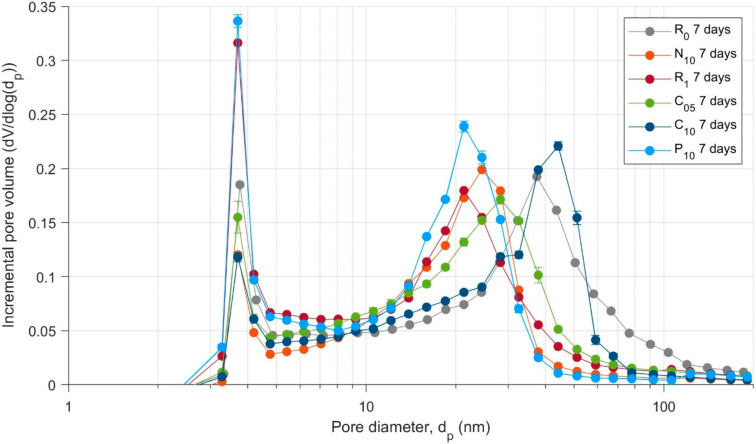
Pore size distributions for mixes aged 7 days, with error bars showing ± 1 s.d. of the results.

**Table 1 materials-13-04097-t001:** Chemical properties and loss on ignition of the Australian type general purpose (GP) cement, complying with AS 3972–2010 [32].

Constituent ^1^	SiO_2_	Al_2_O_3_	Fe_2_O_3_	MgO	CaO	Na_2_O	TiO_2_	K_2_O	SO_3_	LOI
OPC (%)	19.90	4.70	3.38	1.30	63.93	0.17	0.245	0.446	2.54	2.97

^1^ Trace constituents MnO: 0.079% and P_2_O_3_: 0.063%.

**Table 2 materials-13-04097-t002:** Mix proportions for the specimen types investigated. Each mix had a w:b of 0.4.

Parameter	R_0_	N_10_	R_1_	C_05_	C_10_	C_25_	P_10_
Cement (g)	741.4	741.4	741.4	741.4	741.4	741.4	741.4
Water (g)	296.6	295.9	296.6	296.2	295.9	294.7	295.9
CNT dose (wt.% OPC)	-	0.1	-	0.05	0.1	0.25	0.1
Superplasticiser (wt.% OPC)	-	-	0.275 ^a^	0.2	0.4	1.0	0.4
Ultrasonication (20 kJ)	N	Y	Y	Y	Y	Y	Y ^b^

^a^ Superplasticiser dose for R_1_ was selected to provide a similar mini slump diameter to that of specimen C_10_. ^b^ Superplasticiser and water mixture was ultrasonicated, with CNTs added to the cement paste after mixing.

**Table 3 materials-13-04097-t003:** Comparison of composition for VCCTL Cement 116 and Australian type GP cement.

Composition and Mix Proportions		Mass Fraction (%)	
		Cement 116	GP Cement
OPC Principal Phases	C_3_S	66.86	61.3
	C_2_S	22.41	22.2
	C_3_A	6.596	4.6
	C_4_AF	4.134	3.4
Binder		71.43	71.43
Water		28.57	28.57

**Table 4 materials-13-04097-t004:** Experimental water permeability coefficients (K), with the 95% confidence interval widths in parentheses.

Specimen	Water Permeability Coefficient × 10^–13^ m/s
	Aged 7 Days
R_0_: Reference (w/out super)	11.18 (0.09)
N_10_: 0.1% CNTs (no super)	2.67 (0.02)
R_1_: Reference (w/super)	2.22 (0.02)
C_05_: 0.05% CNTs	6.25 (0.05)
C_10_: 0.1% CNTs	10.03 (0.08)
C_25_: 0.25% CNTs	7.80 (0.06)
P_10_: 0.1% CNTs (agglomerated)	10.62 (0.08)

**Table 5 materials-13-04097-t005:** Mean (±1 s.d.) calculated hydraulic conductivity coefficients from permeability simulations on 100 µm x 100 µm x 100 µm cube subvolumes.

×10^–8^ m/s	R_1_ ^a^ 7 Days	C_10_ ^a^ 7 Days	C_10_ ^a^ 28 Days	V_1_ ^b^ 7 Days	V_1_ ^b^ 28 Days
X direction	3.3 (0.5)	10.9 (3.3)	4.9 (0.4)	17.8 (1.0)	15.1 (1.1)
Y direction	3.3 (0.8)	11.7 (3.8)	4.7 (0.4)	17.7 (1.7)	14.9 (1.7)
Z direction	3.0 (0.2)	10.8 (4.1)	4.9 (0.7)	17.7 (0.8)	14.9 (1.1)
Mean	3.2	11.1	4.8	17.7	15.0

^a^ Reconstructed microstructure from X-ray microtomography; ^b^ Simulated microstructure from hydration simulation using VCCTL software.

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
