# Peer review of "Durability Characterisation of Portland Cement–Carbon Nanotube Nanocomposites"

_materials, 2020, doi:10.3390/ma13184097_

Round 1

Reviewer 1 Report

The manuscript mainly deals with an experimental investigation on the durability of CNTs-cement composites along with computer simulation. A series of test on chloride diffusivity, water sorptivity, water permeability and porosity, together with flow and fluid simulations are presented. This is a topic that has not been widely covered in the literature, therefore, the manuscript is somehow interesting and well written. Here are some issues that can help to improve its quality before the manuscript is accepted.

DETAILED COMMENTS

  • Line 70-71: Could you please be more precise saying “the durability enhancements were attributed to the microstructural changes resulting from the admixed CNTs”. Please, could you provide more information about the effects of CNTs on the microstructure of cement-based composites?
  • Line 73-79: The accuracy of rapid chloride ion penetrability test method (equivalent to ASTM C1202) is associated with the electrical resistivity and conductivity of the materials. When CNTs is added to the system, the electrical properties of the composite material will suffer from some changes. How the changes affect the test method?
  • Line 91-94: The conflicting results pertain to the effects of CNTs upon chloride diffusion in terms of different test methods should be explained.
  • Line 147: In Section 2.1, a SEM image of CNTs should be added.
  • Line 158: Table 1 is out of the scope of page layout.
  • Line 199-202: Please explain why smaller specimens were used in water sorptivity test.
  • Line 284-359: It is inappropriate that Subsection 2.3.6 and 2.3.7 are included in Section 2.3 because the both subsections refer to computer simulations rather than physical experiments. Moreover, results from the simulations are presented and discussed in these subsections. I suggest that the results obtained should be moved to “Results and discussion” section. Also, a separate section should be added to exclusively describe the methodology of computer simulations.
  • Line 468-496: It seems to me that the simulated cement hydration, microstructure and fluid flow is little correlated to the experimental results. Please, if possible, explain the relationship between the simulated and physical experimental results, especially on the permeability of OPC-CNTs composites. Additionally, maybe the readers wonder that how the action of CNTs is considered when the simulation is developed and performed.
  • Line 444-451: The layout is irregular. Please check it.
  • Line 299-301: Why 1 µm separation is selected to consider the effects of the added superplasticiser in hydration simulation? Please explain it properly.
  • Line 568-599: The conclusion section is more like only experimental results. It is too lengthy and exhaustive. Please simplify and improve this section.
  • Line 600-602: The sentences should be reconstructed. Please check them carefully.
  • Line 617: “Grant N°: LE13010006” should be changed into “Grant No: LE13010006”.
  • Lines 621-749: Reference list does not conform the requirements of the journal.

Reviewer 2 Report

This paper focused on the effects of superplasticizer, CNT dosage and CNT dispersion quality on the durability of CNT-cement composites. Durability including steady-state chloride diffusion, uptake of water by sorption, flow of permeable water acting under high water pressure is addressed. It is meaningful work for understanding CNT-cement composites and applying them in the field. The following questions are worth the author's consideration:

  1. In Fig.8, where are CNTs, and where are C-S-Hs or AFts?
  2. The authors thought that the CNT dosage was more influential for the durability characteristics of the cementitious nanocomposites than poor CNT dispersion, what is the recommended optimal CNT dosage?
  3. “The results of 9-month steady-state chloride diffusion tests showed that, contrary to the relative performance deterioration observed in the water sorptivity and permeability, the coefficient of chloride diffusion was reduced by up to 63% with the addition of superplasticiser-dispersed CNTs at a dosage of 0.05 to 0.1 wt.%.” Is there any influence on the age of the sample during the test? 9 months versus 7 days versus 28 days?
  4. “A beneficial effect is observed upon the permeability performance as the nanocomposite ages; specimen C10 28 days exhibited a 30% lower average hydraulic conductivity than the 7-day C10 specimen.” According to Table 5, it is not 30%.
  5. There is a big gap between the results of the simulation and those of the experiment.

Reviewer 3 Report

The present paper reports the results of an extensive experimental and theoretical study aimed at investigating the durability performance of cementitious composites incorportating carbon nanotubes.

The manuscript is well written and organized. In the introduction section, the Authors reported a comprehensive state of the art which also well places their contribution. The methods are rigorous and the results are well analyzed.

I only suggest to consider the feasibility of dedicating a separate section to the numerical simulations instead of including it on the experimental methods and results.
